

# Leptin administration does not influence migratory behaviour in white-throated sparrows (*Zonotrichia albicollis*)

Emma Churchman[1] and Scott A. MacDougall-Shackleton[1,2,3]

[1] Department of Biology, University of Western Ontario, London, Ontario, Canada
[2] Department of Psychology, University of Western Ontario, London, Ontario, Canada
[3] Advanced Facility for Avian Research, University of Western Ontario, London, Ontario, Canada

## ABSTRACT

Migratory flights by birds are among the most energetically demanding forms of animal movement, and are primarily fueled by fat as an energy source. Leptin is a critical fat-regulation hormone associated with energy balance in non-avian species but its function in birds is highly controversial. Prior research indicated the effects of leptin differed between birds in migratory condition or not, but no research has assessed the effect of leptin on migratory behaviour itself. In this study, our objective was to determine if leptin affects migratory restlessness and fat deposition in migratory songbirds. We used photoperiod manipulation to induce spring migratory condition, and measured migratory restlessness in leptin-injected and saline-injected white-throated sparrows (*Zonotrichia albicollis*). Leptin treatment had no effect on migratory restlessness nor fat deposition, providing evidence that leptin does not influence avian migratory motivation or behaviour. Our results also further support the idea that birds in a hyperphagic migratory condition may be insensitive to leptin.

# INTRODUCTION

The migratory flights of birds are amongst the most energetically demanding activities performed by animals and can involve extended periods of high-intensity exercise with little to no food intake. Birds use fat as their primary metabolic fuel and deposit large amounts of fat in preparation for migration (*Dingle, 1996*; *Jenni & Jenni-Eiermann, 1998*; *Singh et al., 2018*). Depleted fat reserves, and the time it takes to refuel, are major determinants in how long birds remain at stopover sites between bouts of migratory flight (*Eikenaar & Schlafke, 2013*; *Lupi, Slezacek & Fusani, 2019*; *Goymann et al., 2010*). Thus, the regulation of stored body fat is critical to successful migration. Fat reserves and refuelling are controlled by stimulatory and inhibitory circuits in the hypothalamus (*Boswell, 2005*) under the influence of hormones that signal body condition (*Boswell & Dunn, 2015*). Two mammalian fat regulation hormones, ghrelin and adiponectin, have been proposed to regulate fat and influence bird migration, including the timing of departure from stopover sites (*Goymann et al., 2017*; *Stuber et al., 2013*). However, leptin, an important appetite-regulating hormone in mammals sometimes referred to as the master energy balance hormone, has been less studied with respect to avian migration.

Corresponding author
Emma Churchman,
echurchm@uwo.ca

In mammals, leptin is secreted from adipose tissue and binds with receptors in the hypothalamus to signal fat deposition. As well, circulating leptin is directly proportional to fat levels (*Considine et al., 1996*). Thus, with the emergence of leptin's integral role in controlling mammalian fat and energy balance, it is of high interest to determine if it plays a similar role in birds.

Leptin was characterized in birds almost two decades after it was in mammals. This may be attributed to extreme guanine-cytosine content, occasionally referred to as the "dark side of the genome", and low expression levels (*Friedman-Einat & Seroussi, 2019*). Technological advances in next-generation RNA sequencing resulted in the successful identification of the avian leptin gene (*Huang et al., 2014*; *Friedman-Einat et al., 2014*; *Prokop et al., 2014*). This identification indicated the pattern of genes surrounding the avian leptin gene to be syntenic with those in the mammalian genome. Structural modelling of this avian leptin ortholog revealed a highly conserved hydrophobic core (*Prokop et al., 2014*). However, an early erroneous leptin sequence identified prior to the emergence of high-depth genome sequencing resulted in over 100 papers published, many of which contradict results obtained in those using the correct leptin sequence (*Friedman-Einat & Seroussi, 2019*). The combination of the delay in discovering a genuine sequence of the avian leptin gene and potentially misleading published papers using an erroneous sequence has resulted in a distinct gap in knowledge on the function of leptin in birds.

In theory, avian leptin could be important in the control and success of migration by regulating when birds initiate migration or leave a stopover site based on fat accumulation. However, this mechanism would require avian leptin to function in the same way as its mammalian ortholog. Previous studies have produced conflicting results regarding whether avian leptin correlates with fat reserves and signals this to the brain. *Kuo et al. (2005)* observed leptin administration resulted in decreased food intake in chickens, and suggested avian leptin may function similarly to its non-avian counterparts. *Kordonowy, McMurtry & Williams (2010)* found leptin acted differently in breeding *vs.* non-breeding birds, and suggested leptin function depends on the breeding state of the bird. *Gogga et al. (2013)* suggested that leptin does not reflect accumulated fat in migratory birds and is therefore not involved in energy balance, nor as a control of body fat. *Cerasale, Zajac & Guglielmo (2011)* found leptin affected food intake and fat accumulation in wintering, but not migratory sparrows. Thus, while several studies have examined the effects of leptin treatment on birds, there is no consensus as to its function. To add to the complexity of the debate, it appears that leptin's function may vary depending on whether or not a bird is in a hyperphagic migratory state (*Cerasale, Zajac & Guglielmo, 2011*; *Zajac et al., 2011*). However, there is no information describing the effect of leptin on migratory behaviour itself.

Although mammalian and avian leptin have different structures, several studies have found effects of mammalian leptin on bird behavior. Human (*Kuo et al., 2005*) and murine (*Cerasale, Zajac & Guglielmo, 2011*) leptin affected feeding behaviour in poultry and songbirds, respectively. Murine leptin also affected food-caching behavior in titmice (*Henderson et al., 2018*). These studies suggest that despite the important differences between mammalian and avian leptin that there may be commonalities.

Migratory behaviour can be assessed in captive songbirds using a proxy for the behavioural changes a wild bird would exhibit. Captive birds demonstrate a behaviour called nocturnal migratory restlessness (or *Zugunruhe*), that can include wing whirring and hopping from perch to perch (*Agatsuma & Ramenofsky, 2006*). The intensity and persistence of migratory restlessness is positively correlated with the probability of birds leaving a site to migrate if they were in the wild (*Wingfield, Schwabl & Mattocks, 1990*; *Eikenaar et al., 2014*). Migratory restlessness is useful to determine migratory condition in captive birds and many studies have found a positive relationship between accumulation of fuel (fat) reserves and migratory condition (*Fusani et al., 2009*; *Eikenaar & Schlafke, 2013*; *Lupi et al., 2016*). Prior research has demonstrated that increased levels of appetite-regulating hormones ghrelin and adiponectin both increase migratory restlessness (*Goymann et al., 2017*; *Stuber et al., 2013*). This provides further evidence for the role of energy balance hormones in avian fat regulation and migration, and that migratory restlessness can be used as a proxy to assess their role in migratory behaviour.

In this study, we induced migratory condition in white-throated sparrows (*Zonotrichia albicollis*) with photoperiod manipulation. We administered leptin through a series of injections and used infrared video analysis to measure nocturnal migratory restlessness. We predicted that if leptin is a lipostatic signal in songbirds then sparrows injected with leptin should increase their migratory restlessness. We also measured body composition to assess if leptin affected body fat deposition as the sparrows came into a migratory state. We predicted that sparrows injected with leptin should exhibit higher levels of migratory restlessness, but might potentially deposit less fat if leptin induces negative feedback.

## MATERIALS AND METHODS

### Pre-experiment protocol

White-throated sparrows (*Zonotrichia albicollis*) are an abundant short- to medium-distance migrant North American songbird whose migratory restlessness behaviour has been well-characterized, and whose response to leptin has been previously studied (*Cerasale, Zajac & Guglielmo, 2011*; *Zajac et al., 2011*). We captured 24 white-throated sparrows near Long Point, Ontario, during fall migration in October 2017 (See Fig. 1 for experimental timeline). Sparrows were held under a scientific collection permit from the Canadian Wildlife Service (CA-0244) and provided with *ad libitum* access to food and water. Food was a 50/50 mix of Mazuri Small Bird Maintenance (PMI Nutrition International, Brentwood, MO, USA) and Living World Premium Seed Budgie Mix. The University of Western Ontario's Animal Care Committee approved all animal procedures (protocol #2015-055).

We held the sparrows in outdoor aviaries at the University of Western Ontario's Advanced Facility for Avian Research until November 2017, at which point we moved them to indoor environmental chambers in individual cages (38 cm × 38 cm × 38 cm) at 16 °C. Upon bringing sparrows inside the facility in November 2017, we initially maintained a photoperiod that mimicked natural conditions.

## Experimental Procedure

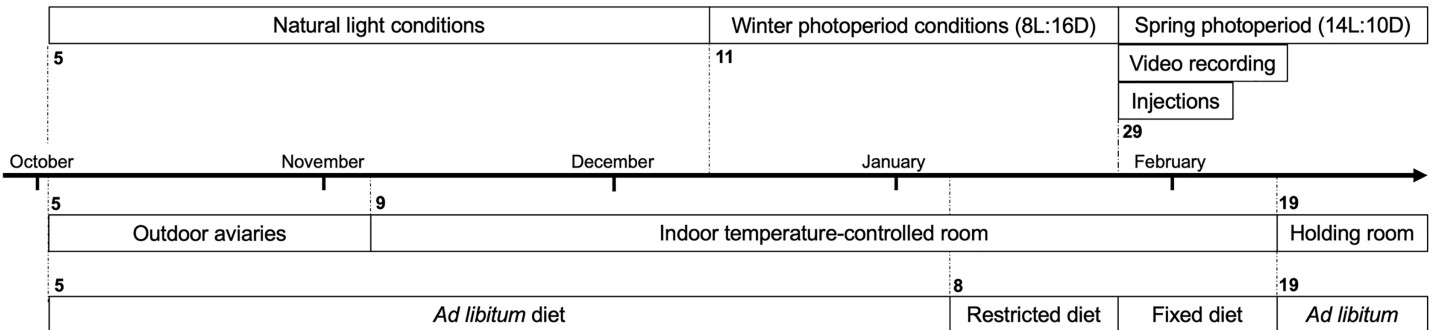

## Bird Care

**Figure 1 Timeline of experimental procedure and bird care for white-throated sparrows (*Zonotrichia albicollis*).** The top portion indicates key dates associated with experimental injection procedure of either leptin or saline, and nocturnal video recording. The bottom portion indicates key dates associated with bird care including housing and diet. Bolded numbers indicate the date the phase started on.

White-throated sparrows are a dimorphic species and are classified as either white-stripe or tan-stripe morphs (*Falls & Kopachena, 1994*). Sex and morph are both variables that can affect migratory behaviour (*Kelly et al., 2020*) in white-throated sparrows and so were determined prior to experimentation to balance treatment groups. We collected blood samples of approximately 20 μL *via* brachial venipuncture that was blotted to sterile filter paper and allowed to dry. We then extracted DNA and amplified regions to determine sex and morph using previously published protocols (*e.g.*, *Kelly et al., 2020*).

Six weeks prior to the experiment, we adjusted the photoperiod to simulate short winter days (8L:16D). We then limited the diet of the sparrows for 2 weeks prior to the beginning of the experiment to lean the sparrows down to a more realistic pre-migratory condition, and to control for individual leptin concentration levels prior to experimental manipulation. During the experiment, sparrows remained on a fixed diet, but were not food restricted, to control for variation in body condition due to variation in food availability.

### Experimental protocol

We measured the amount of fat on each sparrow the day before the experiment began (day 0) using a Quantitative Magnetic Resonance (QMR) scanner (Echo MRI-B, Echo-Medical Systems, Houston, TX, USA). This scanner measures fat non-invasively with ±11% accuracy (*Guglielmo et al., 2011*). On 29 January 2018, we pseudorandomly assigned each sparrow to a treatment group (leptin or saline), balancing for fat deposition, sex, and morph. On this date (Day 1 of the experimental period) we photostimulated the sparrows to induce spring migratory behaviour by adjusting the light schedule to mimic the long days of spring (14L:10D) and began a 2-week series of daily injections. This photoperiod

manipulation reliably induces spring migratory restlessness in this species (*Cerasale, Zajac & Guglielmo, 2011*; *Zajac et al., 2011*; *Vandermeer, 2013*). Following prior studies (*Cerasale, Zajac & Guglielmo, 2011*; *Zajac et al., 2011*), we gave each sparrow an intramuscular injection to the pectoral muscle twice per day at lights on and 6 h after, using 31-gauge insulin needles. Sparrows in the control group received 0.01 mL PBS per injection (1 µg/g body mass; 1 mM phosphate buffer saline, pH ~7.4; Bio Basic Inc., Markham, Ontario, Canada). Sparrows in the experimental group received 0.01 mL murine leptin per injection (1 µg/g body mass; murine leptin, Shenandoah Biotechnology Inc., Warwick, PA, USA, purity >95% as measured by RP-HPLC). This dose and treatment has been proven in previous studies assessing behaviour in sparrows (*Cerasale, Zajac & Guglielmo, 2011*; *Zajac et al., 2011*). Each bird received the injection in the left pectoral muscle in the morning, and right pectoral muscle in the afternoon. We swabbed the injection site with 70% ethanol to clear feathers from the site before injection.

We video recorded the activity of each sparrow from lights off to lights on (10 h per bird) every night from Day 1 of injections to Day 14, and then for a week after cessation of injections (Day 21). Infrared (850 nm) lights (Smart B-Series, model AT-35-B) illuminated the sparrows at night. The infrared lights and video cameras (Black and White High Res Ex View Micro Video, Super Circuits, Austin, TX, USA) were secured to a metal shelving unit facing the sparrows and connected by wires to two computers set up outside the room to save the recordings in real time each night. The day after the video recordings were completed (Day 22), we measured the fat of each sparrow again to assess changes in fat as the sparrows came into a migratory state.

We used Noldus EthoVision XT behavioural software (Noldus Information Technology, Wageningen, GE, NL, www.noldus.com) to quantify the nocturnal migratory restlessness. We created a reference image by editing out the sparrow from a screenshot of each video to be analyzed using Pixlr (*PIXLR Materials, 2008*). EthoVision used the reference image to obtain most accurate noise reduction, and to track movement of the bird that differed from background. Each night recorded as a single trial starting when lights went off and ending the moment before lights came back on (10 h per night). EthoVision measured the time a sparrow spent mobile (s), by quantifying body area fluctuations. We used this time spent mobile as an indicator of migratory restlessness. We also watched the videos to ensure EthoVision accurately measured the bird's behaviour and to confirm the birds displayed nocturnal behaviours characteristic of migratory restlessness: wing whirring and hopping from perch to perch instead of sleeping.

## Statistical analysis

We analyzed the amount of time spent mobile at night (hereafter migratory restlessness) using linear mixed models (LMM) in *RStudio Team (2015)* (v1.1.456). We used migratory restlessness as the dependent variable. This data was heavily-right skewed, so we square-root transformed the data prior to further analysis. Candidate models differed in combinations of trial date, treatment, sex, morph, and trial*treatment. Sex compared male *vs.* female sparrows, while morph compared white-stripe *vs.* tan-stripe plumage

morphs. We ran models to assess the effect of leptin on the total experimental period, as well as models focusing on migratory restlessness within the injection *vs.* post-injection recovery phases. All models included a random effect of trial*bird ID to account for non-linear increases in standard error over time. Initially, we constructed 27 candidate models to assess the effect of leptin on migratory restlessness over the duration of the entire experiment (Days 1–21), including both the injection and post-injection recovery phase. We then ranked the candidate models based on AICc value. We averaged the top models that were separated by a difference of less than two second-order Akaike's information criterion (AICc). To test for effects of leptin individually in each phase we constructed 17 models separately in each phase: injection (Day 1–14) and recovery (Day 15–21). For both phases, we averaged the top most supported models that were separated by a difference of less than two AICc. An averaged model is a weighted average of parameter estimates where the weights are calculated based on the ΔAIC(c) scores. Our model used conditional averaging, where the slope for the averaged parameters is based only on models that include that parameter. In contrast, full averaging uses a partial slope of 0 for models that do not have a value for a particular parameter. We assessed validity by assessing normality, linearity and equal variance, and leverage. Each model fulfilled the assumptions to be an appropriate model for these data. We assessed changes in body fat composition using a two-way repeated measures ANOVA in IBM SPSS Statistics 27 (SPSS Inc, 2009). We used body fat as the dependent variable with time and treatment as the independent variables.

## RESULTS

### Migratory restlessness

Of the candidate models we constructed to assess migratory restlessness in response to leptin administration over the full experimental period (21 days), the averaged top model included the effects of trial, morph, and phase. There was a significant effect of trial date on migratory restlessness ($z = 3.27$, $p = 0.001$; Fig. 2). The parameter estimates for morph ($z = 1.80$, $p = 0.060$) and phase ($z = 0.575$, $p = 0.565$) were not significantly different from zero. Treatment with leptin was included in the fifth most supported model, but did not have a significant effect on migratory restlessness ($t = -0.196$, $p = 0.845$; Table 1).

Of the candidate models we constructed to assess migratory restlessness in response to leptin administration during the injection phase (day 1–14), the averaged top model included the effects of trial, morph, and treatment. There was a significant effect of trial date ($z = 3.40$, $p < 0.001$; Table 2) and morph on migratory restlessness ($z = 1.96$, $p = 0.038$). Tan-stripe morph sparrows exhibited higher activity levels than white-stripe morph sparrows for the duration of the experiment. Treatment was included in the averaged top model but the parameter estimate did not significantly differ from zero ($z = 0.488$, $p = 0.623$). In the recovery phase, the single most supported model included the effect of trial date, treatment, morph, and the interaction between trial date and treatment. There was a significant effect of trial date ($z = 2.64$, $p = 0.019$; Table 3). There was no significant effect of treatment ($z = 1.79$, $p = 0.073$), the interaction between trial date and treatment ($z = 1.69$, $p = 0.091$), nor morph ($z = 0.96$, $p = 0.357$).

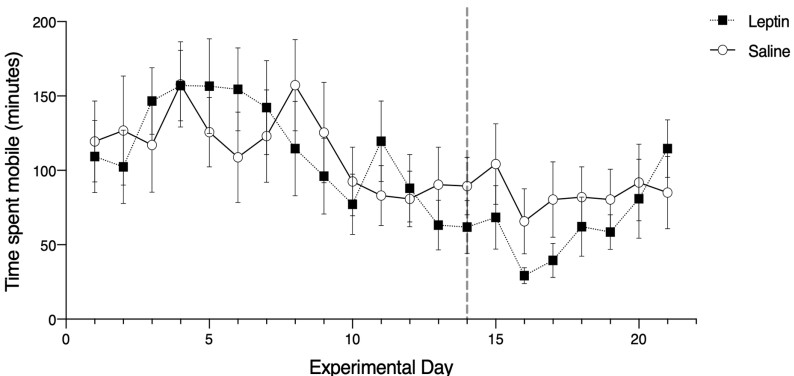

**Figure 2 Effects of leptin treatment on migratory restlessness, measured as time spent mobile (minutes), (mean ± SEM) in white-throated sparrows (*Zonotrichia albicollis*).** Sparrows in the experimental group received leptin injections and are denoted by the black squares. Sparrows in the control group received saline injections and are denoted by the open circles. The vertical dotted line indicates the cessation of injections. There was no significant effect of leptin on migratory restlessness over time.

**Table 1 Selection of models predicting captive white-throated sparrow migratory restlessness during the full 3 week experiment, including both injection and recovery phases.**

|  | AICc |  |  | ΔAICc | $w_i$ |  |
|---|---|---|---|---|---|---|
| **A. Ranked candidate models:** |  |  |  |  |  |  |
| Trial + Treatment + Morph | 4,620.0 |  |  | 0.00 | 0.203 |  |
| Trial + Morph + Phase | 4,621.1 |  |  | 1.15 | 0.114 |  |
| Trial + Morph | 4,621.7 |  |  | 1.74 | 0.085 |  |
| Null | 4635.2 |  |  | 15.24 | 0.000 |  |
|  |  | Estimate | SE | 95% CI |  | *p*-value |
| **B. Parameter estimates of averaged top models:** |  |  |  |  |  |  |
| Intercept |  | 80.31 | 7.19 | [*66.19–94.42*] |  |  |
| Trial |  | −1.51 | 0.35 | [*−2.20 to −0.82*] |  | <0.001 |
| Morph (tan stripe) |  | 16.38 | 8.70 | [−0.71 to 33.48] |  | 0.060 |
| Phase (post) |  | −2.47 | 4.28 | [−10.89 to 5.95] |  | 0.565 |

Note:
(A) Top 3 candidate linear models (and null model) predicting the change in migratory restlessness of captive white-throated sparrows after a 2 week experimental injection phase (leptin or saline), followed by a 1 week post-injection phase. Reported in the table is the second-order Akaike's information criterion (AICc), difference in AICc between candidate models ( AICc), and proportional weight of each model (wi). (B) Real function parameters of the best-fitting model predicting migratory restlessness of white-throated sparrows injected with leptin or saline. A higher estimate indicates higher levels of migratory restlessness. Italics indicate traits for which the 95% confidence interval (CI) surrounding the estimate does not overlap with zero.

## Body fat

There was no significant difference in fat between control and leptin injected birds over the course of the experiment ($F_{1,10} = 0.803$, $p = 0.375$; Fig. 3). Time did not have a significant effect on fat deposition ($F_{1,10} = 0.046$, $p = 0.831$), nor did treatment ($F_{1,10} = 3.157$, $p = 0.083$).

**Table 2 Selection of models predicting captive white-throated sparrow migratory restlessness during experimental injection phase.**

| | AICc | | | ΔAICc | $w_i$ | |
|---|---|---|---|---|---|---|
| A. Ranked candidate models: | | | | | | |
| Trial + Morph | 3,150.7 | | | 0.00 | 0.303 | |
| Trial | 3,152.6 | | | 1.84 | 0.121 | |
| Trial + Treatment + Morph | 3,152.6 | | | 1.87 | 0.119 | |
| Null | 3,160.2 | | | 9.50 | 0.003 | |
| | | Estimate | SE | 95% CI | | p-value |
| B. Parameter estimates of averaged top models: | | | | | | |
| Intercept | | 82.10 | 7.94 | [66.48–97.72] | | |
| Trial | | −1.83 | 0.52 | [−2.86 to −0.79] | | <0.001 |
| Morph (tan stripe) | | 18.77 | 9.03 | [1.01–36.52] | | 0.038 |
| Treatment (saline) | | −4.39 | 8.99 | [−22.07 to 13.29] | | 0.623 |

Note:
(A) Top 3 candidate linear models (and null model) predicting the change in migratory restlessness of captive white-throated sparrows after a 2 week experimental injection phase (leptin or saline). Reported in the table is the second-order Akaike's information criterion (AICc), difference in AICc between candidate models ( AICc), and proportional weight of each model ($w_i$). (B) Real function parameters of the best-fitting model predicting migratory restlessness of white-throated sparrows injected with leptin or saline. A higher estimate indicates higher levels of migratory restlessness. Italics indicate traits for which the 95% confidence interval (CI) surrounding the estimate does not overlap with zero.

**Table 3 Selection of models predicting captive white-throated sparrow migratory restlessness during post-injection recovery phase.**

| | AICc | | | ΔAICc | $w_i$ | |
|---|---|---|---|---|---|---|
| A. Ranked candidate models: | | | | | | |
| Trial | 1,454.3 | | | 0.00 | 0.208 | |
| Treatment + Trial + Treatment × Trial | 1,455.4 | | | 1.04 | 0.123 | |
| Trial + Morph | 1,455.7 | | | 1.38 | 0.104 | |
| Null | 1,456.0 | | | 1.64 | 0.092 | |
| Trial + Treatment | 1,456.1 | | | 1.80 | 0.084 | |
| | | Estimate | SE | 95% CI | | p-value |
| B. Parameter estimates of averaged top models: | | | | | | |
| Intercept | | −17.11 | 31.57 | [−79.43 to 45.20] | | |
| Trial | | 3.99 | 1.69 | [0.65–7.32] | | 0.019 |
| Treatment (saline) | | 77.46 | 42.90 | [−7.31 to 162.22] | | 0.073 |
| Trial × Treatment (saline) | | −3.88 | 2.28 | [−8.38 to 0.62] | | 0.091 |
| Morph (tan stripe) | | 9.95 | 10.71 | [−11.22 to 31.12] | | 0.357 |

Note:
(A) Top 4 candidate linear models (and null model) predicting the change in migratory restlessness of captive white-throated sparrows after a 2 week experimental injection phase (leptin or saline), followed by a 1 week post-injection phase. Reported in the table is the second-order Akaike's information criterion (AICc), difference in AICc between candidate models ( AICc), and proportional weight of each model (wi). (B) Real function parameters of the best-fitting model predicting migratory restlessness of white-throated sparrows injected with leptin or saline. A higher estimate indicates higher levels of migratory restlessness. Italics indicate traits for which the 95% confidence interval (CI) surrounding the estimate does not overlap with zero.

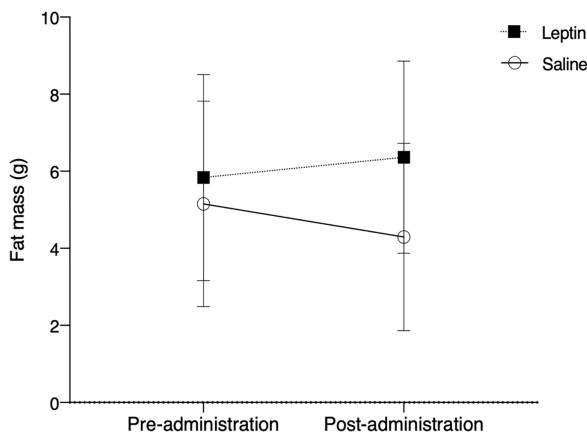

**Figure 3** **Fat mass (mean ± SEM) of white throated sparrows (*Zonotrichia albicollis*) before (Day 0) and after experimental injections (Day 22).** Sparrows in the experimental group received leptin injections and are denoted by the black squares. Sparrows in the control group received saline injections and are denoted by the open circles. The vertical dotted line indicates the cessation of injections (day 14). There was no significant effect of leptin on migratory restlessness over time.

## DISCUSSION

The endocrine system is critical to the initiation and mediation of avian migration. Hormones like corticosterone and testosterone have been proven to influence migratory activity (*Eikenaar et al., 2018*; *Tonra, Marra & Holberton, 2011*; *Vandermeer, 2013*). The role of leptin however, is less clear. The main objective our study was to determine if leptin directly influences the onset of migratory activity in songbirds. While leptin was discovered and characterized in mammals over two decades ago (*Zhang et al., 1994*), sequencing avian leptin proved to be much more difficult and contentious (*Friedman-Einat & Seroussi, 2019*). Following the successful identification of an avian leptin ortholog, it then became important to understand the function of the hormone. Mammalian leptin's primary function is to signal fat deposition, and fat deposition is crucial to avian migration initiation. Thus, the idea that avian leptin may influence migratory behaviour is a plausible hypothesis (*Cerasale, Zajac & Guglielmo, 2011*; *Gogga et al., 2013*).

If leptin were to act analogously to non-avian leptin and signal fat deposition, then we expected an increase in disposition to migrate in birds injected with leptin. However, we found no evidence that administered leptin affected migratory disposition. Leptin injection did not increase migratory restlessness, suggesting that increased leptin is not involved in the onset of migratory behaviour in white-throated sparrows. We also found no effect of increased leptin on fat deposition over the duration of our experiment, which we expected if leptin is a lipostatic signal. In particular we expected lower fat levels if leptin administration falsely signalled increased fat deposition, which was not consistent with our results. Saline injected birds also did not accumulate significant increases in fat, thus it is important to note our experiment assessed the initial onset of migration. It is possible 21 days after a photoperiod change is not enough time for birds to

deposit significant amounts of fat for migration. It is also possible that the stress of injections or the fixed amount of food provided daily affected fat levels in the birds.

It is possible that our null results could be a result of our use of murine leptin in this study, as avian leptin was not available to synthesize at the time. Avian and mammalian predicted protein sequences show low sequence similarity (~30%). The predicted protein sequence of avian leptin also varies among birds (~45%). However, the structure of avian and mammalian leptin binding-domains are highly conserved, with ~80% identical amino acids (*Hen et al., 2008*; *Adachi et al., 2008*). To our knowledge, no study has compared the effect of endogenous and heterologous leptins (*Friedman-Einat & Seroussi, 2019*). Therefore, for our study, murine leptin injections were the same dose and administration regime as in other studies that found effects on songbird behaviour (*Cerasale, Zajac & Guglielmo, 2011*; *Zajac et al., 2011*). For this reason, it is unlikely that the null results stem from insufficient dosing or heterologous leptins, though further studies would be required to confirm this. It is also possible that leptin treatment closer to nightfall and the onset of migratory restlessness could affect migratory behaviour, but this would require further studies. Our injections during the daytime did not have any detectable direct or indirect effects, but it remains possible that there could be rapid direct effects if birds experienced elevated leptin during the nighttime at the time of migratory behavior.

It is worth noting there was a significant effect of plumage morph during the injection phase of the experiment. Morph has previously been shown to affect migratory activity in white-throated sparrows (*Kelly et al., 2020*). While our study supports this, we also provide evidence that tan-stripe morph sparrows show a higher disposition to initiate migration compared to the white-stripe morph, suggesting that they may advance the onset of spring migration. Although behavioural differences in parenting and territorial singing behaviour between tan-stripe and white-stripe white-throated sparrows are well documented (*Falls & Kopachena, 1994*), our study adds to evidence that indicate these morphs also differ in migration behaviour.

Our results suggest that leptin does not act in a similar role to its non-avian homologues and signal fat deposition in migratory songbirds. This finding conceptually replicates previous work on white-throated sparrows that assessed the role of leptin in foraging, food intake, and fat mass in wintering and migratory birds (*Cerasale, Zajac & Guglielmo, 2011*). *Cerasale, Zajac & Guglielmo (2011)* found that birds decreased foraging behaviour in response to leptin in the winter, but not when in a migratory status. Further, wintering sparrows injected with leptin lost significantly more fat mass, while migratory sparrows injected with leptin actually increased fat mass. Their results indicated that while leptin may influence behaviour and fat deposition in the winter, there is no effect on birds in a migratory condition. *Gogga et al. (2013)* also found that leptin did not reflect the extent of fat in migratory dunlins (*Caldris alpine*). Our study supports the view that migratory birds do not respond to leptin as a hormone that regulates fat, or fat accumulation behaviour.

Unlike migratory birds, wintering and resident birds appear to respond to leptin. Leptin attenuates food intake in non-migratory birds such as chickens, quail, and titmice

(*Denbow et al., 2000*; *Lõhmus et al., 2003*; *Lõhmus, Sundström & Silverin, 2006*). Further, leptin reduced food-hoarding behaviour during winter in coal tits (*Periparus ater*; *Henderson et al., 2018*). The fact that only birds in a non-migratory state respond to leptin administration suggests that birds become insensitive to leptin as they come into a migratory state, at a time when overriding leptin signaling would induce hyperphagia to build critical fat reserves necessary for migration.

An alternative explanation for our null results is that avian leptin is not an adipokine and therefore not expressed in adipose tissue as birds come into a migratory state (*Friedman-Einat & Seroussi, 2019*). Quantitative PCR profiling of leptin mRNA in zebra finches (*Taeniopygia guttata*) indicated that leptin mRNA was expressed in the brain and pituitary, but not in peripheral tissues including fat (*Huang et al., 2014*). However, *Cerasale, Zajac & Guglielmo (2011)* found both leptin receptor isoforms expressed in adipose tissue and the hypothalamus in both wintering and non-wintering birds. Further, they assessed a suppressor of cytokine signalling three (SOCS3), which has been proposed as a mechanism involved in leptin resistance. While their data was not consistent with the expected increase in SOCS3 expression associated with leptin resistance, their experiment sampled 40 days after photoperiod stimulation when birds were already in a migratory state which may have missed shorter term effects. It is possible the changes associated with receptor isoform and SOCS3 expression occur transiently after a change in photoperiod. Our experiment allowed us to assess this initial change in migratory behaviour after a change in photoperiod when birds begin to develop a migratory phenotype. Based on our null results in response to leptin, we therefore suggest as birds come into a migratory state, they may not express leptin, nor the receptors, in fat tissues. Furthermore, SOCS3 may increase at the initial onset of the migratory phenotype to assist in leptin resistance to induce hyperphagia. Future research in this area should consider this possibility and further study the regulation of leptin and leptin receptor expression at the onset of migration.

## CONCLUSIONS

Our study adds to a growing body of research on the hormonal regulation of avian migration. Prior research suggests that avian leptin does not influence the feeding or foraging behaviour of birds preparing for migration. Here, we provide empirical evidence that increased leptin does not influence fat deposition in songbirds, and that leptin does not directly influence migratory behaviour itself. We suggest that migratory birds are insensitive to increases in circulating leptin, or differentially express leptin to facilitate hyperphagia at the onset of the migratory phenotype. Further study of the function of avian leptin is important to better understand the hormonal regulation of migration.

## ACKNOWLEDGEMENTS

We thank B. Sinclair, D. Sherry, S. Lupi, B. Rubin, T. Kelly, and A. Boyer for feedback and advice, along with M. Rebuli and F. Boon for logistic support.

### Funding

This project was supported by the Natural Sciences and Engineering Research Council (NSERC) Canada (Discovery Grant 2018-05658). The funders had no role in study design, data collection and analysis, decision to publish, or preparation of the manuscript.

### Grant Disclosures

The following grant information was disclosed by the authors:
Natural Sciences and Engineering Research Council (NSERC) Canada Discovery Grant: 2018-05658.

### Competing Interests

The authors declare that they have no competing interests.

### Author Contributions

- Emma Churchman conceived and designed the experiments, performed the experiments, analyzed the data, prepared figures and/or tables, authored or reviewed drafts of the article, and approved the final draft.
- Scott A. MacDougall-Shackleton conceived and designed the experiments, authored or reviewed drafts of the article, and approved the final draft.

### Animal Ethics

The following information was supplied relating to ethical approvals (*i.e.*, approving body and any reference numbers):

The University of Western Ontario's Animal Care Committee approved all animal procedures (protocol #2015-055).

### Data Availability

The raw measurements are available in the Supplemental File.

### Supplemental Information

Supplemental information for this article can be found online at http://dx.doi.org/10.7717/peerj.13584#supplemental-information.

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
