# Peer review of "Leptin administration does not influence migratory behaviour in white-throated sparrows (Zonotrichia albicollis)"

_PeerJ, doi:10.7717/peerj.13584_

## Round 0.1 · original submission · Major Revisions

Although all of the points raised by both reviewers need to be fully addressed, I would also like to emphasize the following:

- The discussion needs rewriting to acknowledge the potential shortcomings of the design and to be more circumspect regarding conclusions. The current design does not allow the stated conclusions to be drawn.

- Regarding reviewer 2's point about structural differences in avian and mammalian leptin. Knowing these are very different, what is the rationale behind testing its role in avian physiology?

·

Excellent Review

This review has been rated excellent by staff (in the top 15% of reviews)
EDITOR COMMENT
Thank you for your very considerate review. This is an ideal example of how the review process should work, providing assistance and helpful suggestions to authors to improve their paper.

Basic reporting

The paper is well written, with sufficient, relevant references, and the background information sets up the question sufficiently. Apart from a few editing mistakes (an extra "used" in line 185 and a typo "recordins" in line 168), I could find no other grammatical or spelling errors. The figures are clear, although the figure legend for Figure 3 has text that is clearly copied and pasted from Figure 2, and does not belong in the Figure 3 legend.

The study clearly states the hypothesis and predictions, and the data attempt to address those predictions.

Experimental design

The study is original primary research, the research question is well defined, interesting and meaningful. However, I am not convinced that the way the study was carried out is a decisive test of the stated hypotheses. There are, in fact, two aspects of how the study was set up that might be responsible for the negative (non-significant) results found by the authors, which then in turn does not allow them to draw the strong conclusions they currently draw from the study. These two aspects are:

1) the birds were injected with (murine) leptin twice per day, 6 hours apart, in the light phase. However, the behaviour that was being quantified happens during the dark phase, 8h after the second injection of the day. The authors do not mention anything about the biological half-life of the injected leptin, but I would be surprised if it is still present by the time the behaviour is measured. All the other studies they cite that have done similar things (Henderson et al 2018, Cerasale et al 2011, Zajac et al 2011) quantified behaviours that happened shortly after the injection, not 8-18h later. It is therefor possible that migratory restlessness does respond to leptin, but only acutely.

2) the other prediction was that the birds would gain less fat if injected with leptin. This may suffer from the same problem as above, if the biological half-life is short, as they could easily compensate for the suppressing effect of leptin in the hours after it has worn off, or indeed in the week after the treatments were stopped. There is, however, an even more fundamental problem with the experimental design to test this prediction, and that is that the birds were food restricted for the entire duration of the experiment. I do not understand how the authors expected the birds to gain fat (and indeed, they did not, even though they should when building up to migration), if their access to food was not ad libitum during this period. If your prediction is that the treatment will decrease the fat gain, then setting up the foraging conditions such that fat gain is less likely is not a good test of your hypothesis.

Validity of the findings

At face value, the analyses carried out seem valid and appropriate, and the data collected seem valid as well, although it would have been nice to quantify a little bit how the validation of the EthoVision data came out (e.g. correlation between EthoVision and manual scores for the sample used in the validation). The data contained many zeros and were square root transformed. It is not clear whether after transformation, the data were acceptable for the models. If many zeros were present, there would still be a lower cut-off on the data that is not ideal for linear models.

The raw data are provided for the migratory restlessness, but not for the fat data. These should also be provided (e.g. in a separate tab of the spreadsheet).

The statistical analyses are good, although I would have liked a bit more explanation about how models are averaged. I know this is a common procedure, I just don't know how it's done. Also, I do not understand how Phase and Trial can be in the same analysis together, as they must be highly co-linear. I may not quite understand the nature of the models being fitted, though.

The direct conclusions from the analysis are fine (e.g. there are no significant changes in fat reserves), but the more general conclusions drawn from those findings (e.g. "leptin has not direct influence on avian migratory motivation or behaviour"), are not, due to the design issues outlined earlier. Conclusions about birds being insensitive to leptin are also too far, as even if the experiment had worked as planned, it is testing the effects of EXTRA leptin, on top of what may already be circulating in the bird. So at best they are insensitive to the increases in leptin administered in the experiment.

Additional comments

In the introduction, the authors lay out the history of avian leptin, and the fact that for a long time, the leptin used as "avian" as in fact murine leptin. I think it would be more informative if the authors used "murine leptin" and "avian leptin" throughout the paper, so the reader can follow better. In addition, I would like the authors to justify why they used murine leptin, after setting out in the introduction that "using an erroneous sequence has resulted in a distinct gap in the knowledge on the function on leptin in birds".

Also, if the authors could clearly state which species each study that they cite in the introduction was conducted, that would be very helpful, as "bird" covers 10,000 species.

The statement about the effect of leptin on migratory behaviour (line 84-85) seems to come out of nowhere. Later on the same page, evidence is reviewed that justifies looking for such an effect. I suggest moving that sentence to after the justifying evidence has been reviewed.

The methods could be explained a bit better, especially the "natural" photoperiod the birds were on when being moved to 8L:16D is never stated. Figure 1 is very helpful, because many of those details were unclear from the text. E.g. the text (line 138-140) suggested to me the food restriction was for only 2 weeks, but the figure makes clear it lasted 5 weeks, and covered the entire treatment period and beyond. It is never made clear what the parameters of the food restriction were, and whether they were individual to get a certain fat level or body mass.

I have two minor questions about the tables: in both Table 1.1 and table 1.3, the SE and the 95%CI don't seem to correspond very well for two variables: in Table 1, Treatment has an SE of 0.34 and a 95%CI of -22.07 to 13.28, while in Table 3, Morph has an SE of 0.17, and a 95%CI of -10.35 to 30.09. That seems like quite a mismatch...

Reviewer 2 ·

Excellent Review

This review has been rated excellent by staff (in the top 15% of reviews)
EDITOR COMMENT
Thank you for your thoughtful and thorough review. This is an excellent example of how the review process should serve to constructively improve manuscripts.

Basic reporting

The authors investigated effect of leptin hormone administration in migratory bird (white-throated sparrow; Zonotrichia albicollis) under long day photoperiod (14 hours of light and 10 hours’ dark) during pre-migratory. The objective was to test if leptin administration can induce fat deposition and early appearance of night time migratory restlessness. The manuscript has a few caveats that need to addressed or resolved with caution. Authors have interpreted results and drawn conclusions based on low sample size and limited time points that has been acknowledged in discussion. It is an interesting study but need major revision to improve the presentation and clarity in different sections of the manuscript.

Experimental design

Author acknowledges here that studies on avian leptin has been controversial. I think all the studies in past including this work done in this manuscript uses murine leptin to test its response on avian physiology withought testing how similar murine and avian leptins are in terms of chemical nature, protein sequence, and other biochemical properties.
Are there any reports showing cross reactivity between antibody raised against murine leptin and avian plasma isolate?
My biggest concern is what if avian and mammalian leptin are structurally very different. Knowing it is very different, what is the rationale behind testing its role in avian physiology?

The experimental design is OK and author has mentioned the caveats in discussion.

Validity of the findings

The objective was to test if leptin administration can induce fat deposition and early appearance of night time migratory restlessness.
As expressed earlier, my biggest concern is what if avian and mammalian leptin are structurally very different. Knowing it is very different, what is the rationale behind testing its role in avian physiology?
Experimental design, and statistics used are fine.

Additional comments

Line 42: you can add Singh et al., 2018, Frontiers in physiology showing fat deposition in liver in migratory birds a relevant reference to show how fat is deposited in migratory state and it disappears post-migration.

Line 51: However, leptin ………. balance hormone, has been……migration. The first and half sentence is not connected. Reframe the sentence.

Line 58,59: Break the sentence into two. The message is not clear from the sentence. I guess author wants to mention here that avian leptin could not be sequence earlier due to high GC content and that also makes it challenging to amplify with standard PCR.
The end of the sentence and low expression is not clear here in the sentence.

Line 64: Structural modelling……. Hydrophobic core. What information author want to give here. It doesn’t connect with the previous sentence.

Methods & results-

Line 118: We captured 24 white-throated sparrows. Author used morph in running statistical model and results. It is not clear throughout the method what was the number of different morphs in the analysis.
Also, give power estimates for statistical significance.

Another question: did author corrected individual mass for body size. I think there could be individual body size differences which was not taken into account for body fat.

Line 168: correct recordings spelling in the sentence.

Discussion

Line 227: Hormones like ………..influence migratory activity. I think author should include Mishra et al., 2017 Hormones & Behavior showing role of circulating corticosterone and insulin levels in night time activity migratory (daily levels and rhythm in circulating corticosterone and insulin are altered with photostimulated seasonal states in night-migratory blackheaded buntings).

Adding some information in discussion on structural, biochemical differences on mammalian and avian leptin would help in conjunction with how difference in the structure could be a potential cause of no functional relevance of leptin in birds.

Annotated reviews are not available for download in order to protect the identity of reviewers who chose to remain anonymous.

---

## Round 0.2 · Minor Revisions

Thank you for your attention to the previous comments made by both reviewers in this current revision. Please find attached some additional comments from Reviewer 1 that should be addressed.

·

Basic reporting

The article has much improved from the first draft. I have two more requests to further improve the clarity of the article:

1) Can you please comment in the discussion on whether you've managed to induce a migratory state at all? The lack of change in fat and the DECREASE over time in activity suggest that you may not have. That would change the interpretation a little bit, as part of the interpretation is that birds stop responding to exogenous leptin when in migratory state. Are there other studies that have used photoperiod to induce migratory restlessness in this species and do they see a different pattern?

2) You gave some really nice and clear explanations in the response to reviewers that could also be added (without much effort) to the manuscript itself:
- The explanation of how models are averaged
- The reassurance that after transformation the data met the assumptions of the models

Given the changes made, maybe the title and abstract should talk about "Leptin treatment" rather than just "Leptin"?

Experimental design

This has been properly dealt with.

Validity of the findings

I have no problem with the data or the interpretation, with one exception: the difference in activity between the two morphs is interpreted as a difference in migratory restlessness. Given the doubt that any migratory restlessness was induced and there was no interaction between trial and morph, maybe it's safer to interpret that as difference in activity levels only.

I do also wonder why for the fat data, a "normal" two-way (repeated measures) ANOVA was used, and no the same model selection approach as for the behavioural outcome measures. Sex and morph could affect fat as well, surely? But if there is a good reason, then there is no need to change this.

Finally, in the responses to reviewers, the method of validating the EthoVision data was explained. Please explain that here too. And, if I understand the validation method correctly, can a rank correlation not show this validity?

Additional comments

A few small points:
- why are the tables numbered 1.1, 1.2 and 1.3? Just number them 1, 2 and 3.
- when saying there is a significant effect of morph on activity levels, please tell us which is more active
- line 176 of the Word (tracked) version: "recordings" instead of "recordins"
- line 193 of the Word (tracked) version: please remove "used"

---

## Round 0.3 · accepted · Accept

Thank you for your resubmission and your attentive responses to the reviewers' comments. I very much enjoyed reading this paper and think it makes a welcome addition to the literature.